# Virucidal and Bactericidal Filtration Media from Electrospun Polylactic Acid Nanofibres Capable of Protecting against COVID-19

**DOI:** 10.3390/membranes12060571

**Published:** 2022-05-30

**Authors:** Fabrice Noël Hakan Karabulut, Dhevesh Fomra, Günther Höfler, Naveen Ashok Chand, Gareth Wesley Beckermann

**Affiliations:** NanoLayr Ltd., 59 Mahunga Drive, Mangere Bridge, Auckland 2022, New Zealand; gunther.hofler@nanolayr.com (G.H.); naveen.chand@nanolayr.com (N.A.C.); gareth.beckermann@nanolayr.com (G.W.B.)

**Keywords:** electrospun, nanofibre, polylactid acid, filtration, SARS-CoV-2, filter media

## Abstract

Electrospun nanofibres excel at air filtration owing to diverse filtration mechanisms, thereby outperforming meltblown fibres. In this work, we present an electrospun polylactide acid nanofibre filter media, FilterLayr^TM^ Eco, displaying outstanding bactericidal and virucidal properties using Manuka oil. Given the existing COVID-19 pandemic, face masks are now a mandatory accessory in many countries, and at the same time, they have become a source of environmental pollution. Made by NanoLayr Ltd., FilterLayr^TM^ Eco uses biobased renewable raw materials with products that have end-of-life options for being industrially compostable. Loaded with natural and non-toxic terpenoid from manuka oil, FilterLayr Eco can filter up to 99.9% of 0.1 µm particles and kill >99% of trapped airborne fungi, bacteria, and viruses, including SARS-CoV-2 (Delta variant). In addition, the antimicrobial activity, and the efficacy of the filter media to filtrate particles was shown to remain highly active following several washing cycles, making it a reusable and more environmentally friendly option. The new nanofibre filter media, FilterLayr^TM^ Eco, met the particle filtration efficiency and breathability requirements of the following standards: N95 performance in accordance with NIOSH 42CFR84, level 2 performance in accordance with ASTM F2100, and level 2 filtration efficiency and level 1 breathability in accordance with ASTM F3502. These are globally recognized facemask and respirator standards.

## 1. Introduction

The recent worldwide pandemic, caused by the respiratory disease SARS-CoV-2, has shed light on the importance of air quality and excellent air filtration. Awareness of the impact that air quality has on human health has led to an increase in the demand for better personal protection against airborne particles and microorganisms. Personal protective equipment (PPE) is the most accepted method of self-protection in situations where safe social distancing is not possible. Masks and respirators are undoubtedly the most critical pieces of protective equipment for preventing the spread of particulate matter such as fungi, bacteria, and viruses [1,2,3,4]. However, many concerns remain about the survival of microbes on the face mask surface [5,6,7] re-aerosolisation of settled particles [8], proper management and disposal of old face masks [9], and fomite transmission [10]. In order to impart surface contact killing/deactivation mechanisms in addition to particle filtration, some of these concerns have led to the development of face masks with inherent antimicrobial capabilities. Most face masks contain plastics or similar materials, making their widespread and ever-increasing use a significant environmental issue. In a short span of time, mask usage during COVID-19 has generated millions of tons of plastic waste.

When utilised as a protective filter media layer in facemasks, nanofibres (NFs) have unique properties [11,12,13]. The electrospinning (ES) method is considered to be the best for generating polymeric NFs, as a wide range of synthetic and or bio-based polymers can be used at both the lab and commercial scale [14]. Last year, Beckermann et al. introduced the ability of electrospun PMMA/EVOH NF filter media to pass stringent international standards [11]. However, the authors reported a filter media made of a polymer blend (PMMA/EVOH); materials which are not considered sustainable. Due to the COVID-19 pandemic, the monthly usage of single-use facemasks during 2021 was estimated at 129 billion [15], Ref. [15] resulting in the generation of a lot of waste, much of which inevitably ended up in the environment. Recent trends drive research and development to focus on biodegradability, compostability and reusability of the next generation of face masks.

ES technique can be used to produce functional NFs with enhanced antimicrobial properties when loaded with antimicrobial agents [16,17,18]. Additives such as silver (Ag) [19,20,21,22] and copper (Cu) nanoparticles (NPs) [23] and plant extracts such as *Centella asiatica*, *Myristica andamanica*, aloe vera and grape seed [24,25,26,27,28] have demonstrated effective antimicrobial properties against a wide range of pathogens and can be incorporated into the electrospun formulations. Currently, many antimicrobial textiles are coated with NPs of metals such as copper and silver [29,30,31,32,33,34,35,36]. These NPs provide antimicrobial properties by releasing Ag^+^ or Cu^2+^ metallic ions or reactive oxygen species, triggering bacterial cell-wall damage [37]. These metal species, if loosened from the materials, could leak into grey water (after washing) or onto the consumer’s skin through wearing. Antimicrobial face masks containing copper, silver, or other antimicrobials are widely advertised. The methods for loading antimicrobials onto facemask fibres are often undisclosed, and the risk of metal leaching from these masks is unknown. Recently, Goldfard et al. reported the quantity of silver and copper metals leached from nine commercially available ’antimicrobial’ masks and a 100% cotton face mask used as a control. These masks were exposed to laundry detergent, artificial saliva, and deionized water to mimic normal facemask usage and care. The authors observed that leaching differed significantly depending on the brand, leaching solution and metal NPs. After an hour of exposure, in some cases, the full content of the metals contained in the original face mask leached [38]. Findings such as these have prompted researchers to investigate naturally derived bioactive agents as more favourable alternatives to heavy metals. 

To address both the viral threats and environmental impacts, the development of a unique face mask filter medium is described in this paper, made from electrospun bio-based polylactic acid (PLA). This filter layer boasts an additional antimicrobial functionality imparted by the addition of non-toxic and naturally sourced terpenoids extracted from manuka oil. The antimicrobial properties of manuka triketone have been previously reported in applications such as wound dressing [39] but have not been demonstrated as an active component of face masks. This novel PLA filter media successfully achieved high filtration efficiency for particulate pollutants and pathogens. The PLA ES NF filter media were tested according to the following international standards: ASTM Test Method F2299 [40], ASTM Test Method F3502 [41] and NIOSH 42CFR84 (N95) [42]. Additionally, the antimicrobial performance of this filter media were evaluated against different bacteria strains (*Escherichia coli*, *Staphylococcus aureus*, and *Klebsiella pneumoniae* following the test standard ISO 20743:2013 [43]), viruses (influenza A (H1N1), human coronavirus and SARS-CoV-2 following the test standard ISO 18184:2019 [44]) and a fungus (*Aspergillus niger*) following the test standard ASTM G21 [45]). Lastly, to test the reusability of the filter media after being subjected to 5–10 laundering cycles, the particulate filtration efficiency and antimicrobial activity were tested again according to ASTM F3502 [41] and ISO 20743:2013 [43] standards.

## 2. Experimental

### 2.1. Materials

The following materials have been used to prepare solutions for electrospinning: polylactic acid (PLA); formic acid and acetic acid purchased from Merck (Kenilworth, NJ, USA). Manuka oil was purchased from a local New Zealand supplier. To protect the nanofibre layer, spun-bonded PLA, a non-woven fabric, was used as a substrate and cover layer.

### 2.2. Electrospinning of Nanofibre and Characterisation

The most widely used method to produce polymer NFs is electrospinning. This has been explained in detail by Rutledge and Fridrikh [46]. Nanolayr Ltd. (Auckland, New Zealand) formerly known as Revolution Fibres Ltd., has developed and patented a revolutionary needleless electrospinning process which was used to manufacture this filter media in roll form, and which was used in this investigation. The solutions used in the ES process were made by dissolving a specific amount of polymer into a suitable solvent containing mixtures of formic/acetic acid. The ES process involves applying the polymer solution to positively charged electrodes in an electrostatic field, which causes the polymer solution to draw out and spin into random and continues NFs, which are deposited onto a spun-bonded PLA substrate which is rested on a negatively charged collector plate. Area weights were determined by weighing 100 cm^2^ samples using a Precisa XB220A analytical balance and by dividing the sample mass by the sample area.

The samples of NF were analysed using a scanning electron microscope JEOL JCM-5000. The Fibraquant evaluation software was used to measure average NF diameters from 50 to 100 measurements.

### 2.3. Pressure Drop and Breathing Resistance

The total difference in pressure between two points of a fluid (air) as it flows through the filter media is known as the pressure drop. This drop in pressure is caused by frictional forces which are resisting the flow. The drop in pressure is measured by using TexTest FX 3300 LabAir IV, which complies with EN 14683:2019 + AC:2019 [47].

Breathing resistance or breathability measures the difficulty in inhaling or exhaling through a mask or filter media. This is commonly expressed as a pressure drop across the filter media. PALAS PMFT 1000 in accordance with the ASTM Test Method F3502 [41] and NIOSH 42CFR84 (N95) [42] was used to measure different filter media for breathability.

### 2.4. Filtration Performance Testing 

**Particle Filtration Efficiency:** This procedure was performed to assess the particle filtration efficiency (PFE) of the filter media. Particles of monodispersed polystyrene latex spheres (PSL) were dried, nebulized (atomized), and passed through the filter media and enumerated using a laser particle counter. A one-minute count was performed with the filter media in the system. A control count was performed for one minute without the filter media in the system, before (upstream) and after (downstream) each test article. The filtration efficiency was calculated using the number of particles penetrating the filter media compared to the average of the control values. The air flow rate was maintained at 1 cubic foot per minute (CFM) ±5% during the testing. The procedure used the ASTM F2299 method [40], with some exceptions. While in real world, particles carry a charge, this procedure was carried out in a non-neutralized environment, thus representing a more natural state. The non-neutralized aerosol is also specified in the FDA guidance document on surgical face masks. The % filter penetration (P_filter_) is calculated as the ratio of the particle concentration downstream (C_down_) to the particle concentration upstream (C_up_). The ratio between the particle concentration downstream and the particle concentration upstream is the filter penetration (P_filter_ (%) = C_down_/C_up_ × 100%). PFE is the complement of the filter penetration (PFE (%) = 100% − P_filter_). A PFE of 98% means that the filter media will block 98% of particles (either all particle sizes or specific particle sizes) such that only 2% of particles will pass through the material when air is inhaled or exhaled. The filter media were tested in-house and certified by Nelson Lab, ASTM F2299 [40].

**Bacterial Filtration Efficiency (BFE):** The bacterial filtration efficiency of the material is determined by carrying out a BFE test, which is performed by comparing the bacterial control count upstream of the filter to the bacterial count downstream. A suspension of *Staphylococcus aureus* was aerosolized by a nebulizer and delivered to the test material at a constant rate of flow and air pressure. The challenge delivery was maintained at 1.7–3.0 × 10^3^ colony forming units (CFU) with a mean particle size (MPS) of 3.0 ± 0.3 μm. The aerosols were drawn through a six-stage, viable particle Andersen sampler for collection. This test method complies with ASTM F2101-19 [48] and EN 14683:2019 + AC:2019 [47] standards. PLA filter media were tested and certified by Nelson Lab US.

**Viral Filtration Efficiency (VFE):** The filtration efficiency of the material was determined by performing the VFE test, which is performed by comparing the viral control counts upstream of the filter to the counts downstream.

A suspension of bacteriophage ΦX174 was aerosolized by a nebulizer and delivered to the test material at a constant rate of flow and air pressure. The challenge delivery was maintained at 1.1–3.3 × 10^3^ plaque-forming units (PFU) with a mean particle size (MPS) of 3.0 ± 0.3 µm. Then, the aerosol droplets were drawn through a six-stage, viable particle Andersen sampler for collection. The VFE test procedure was adapted from ASTM F2101 standard. PLA filter media were tested and certified by Nelson Lab US.

The PFE values presented in this paper were measured on the filter media only and non-fabricated masks. The PFE of facemasks can differ during filter testing due to air leakages around the edges of the mask. The PLA NF present within the filter media containing an average areal weight of 2.0 ± 0.1 gsm were tested in-house using a PALAS PMFT 1000 testing system according to NIOSH 42CFR84 (N95) [42], ASTM Test Method F3502 and ASTM Test Method F2299 standards [40]. PLA NF filter media were tested and certified by Nelson Lab to pass NIOSH 42CFR84 (N95) [42], ASTM Test Method F3502 [41] and ASTM Test Method F2299 standards [40]. A comparative summary of the filtration test method requirements for the various international test standards are provided in Table 1.

### 2.5. Electrospinning Solutions 

Electrospinning solutions were prepared by stirring PLA polymer in a mixture of formic acid/acetic acid solvent at room temperature until all the polymers had fully dissolved. Manuka triketone extracted from manuka oil was then added to the solution. 

### 2.6. Antimicrobial Activity

FilterLayr Eco consisting of a functional nanofiber layer sandwiched between two layers of PLA spunbonded nonwoven fabric was tested for antimicrobial activity. The tests followed the protocols outlined in ISO 18184:2019 [44], Determining virucidal activity of textile product, ISO 20743—Determination of antibacterial activity of textile product [43] and ASTM G21—Resistance of Synthetic Polymeric Materials to Fungi [45]. The antimicrobial testing protocol can be found in Appendix A in the supporting information document.

#### 2.6.1. Antibacterial Activity

The PLA filter media were tested against bacterium *Staphylococcus aureus* (ATCC 6538P *Escherichia coli* (ATCC 8739) and *Klebsiella pneumoniae* (ATCC 4352) in accordance with the following test standard ISO 20743:2013—Quantitative antibacterial test on textiles [43]. In addition, the PLA filter media were subjected to 5 and 10 laundering cycles following ISO 6330:2013—Domestic laundering [49], and then tested against Gram-positive bacterium *Staphylococcus aureus* and Gram-negative bacterium *Klebsiella pneumoniae* following ISO 20743:2013—Quantitative antibacterial test method standards [43]. The antibacterial activity rating is shown in Table 2.

Antibacterial Activity Calculation:(1)$$A=(C_{t}-C_{0})-\left(T_{t}-C_{0}\right)=\left(C_{t}-T_{t}\right)$$
where A is the antibacterial activity value, $C_{0}$ is the logarithm average of 3 bacterial colony forming units (cfu) immediately after inoculation of the control specimen, $C_{t}$ is the logarithm average of 3 bacterial colony-forming units (cfu) after specified contact time with the control specimen, and $T_{t}$ is the logarithm average of 3 bacterial colony-forming units (cfu) after specified contact time with the treated specimen. 

Percent reduction calculation:(2)$$\mathrm{Percent}\;\mathrm{Reduction}=1-10^{A}$$

#### 2.6.2. Antiviral Activity

The PLA filter media were tested against three highly potent and prevalent viruses such as Influenza A (H1N1) (ATCC VR-1469), human coronavirus 229E (ATCC VR-740) and SARS-CoV-2 (Delta variant, B.1.617.2; NCBI MZ574052) in accordance with the method outlined in ISO 18184:2019 standard [44]. The antiviral activity rating is shown in Table 3.

Anti-viral activity calculation:(3)$$M_{v}=\log_{10}\left(V_{a}\right)-\log_{10}\left(V_{b}\right)$$
where $M_{v}$ is the antiviral activity value, $\log_{10}\left(V_{a}\right)$ is the logarithm average of 3 infectivity titre values immediately after inoculation of the control specimen, and $\log_{10}\left(V_{b}\right)$ is the logarithm average of 3 infectivity titre values after specified contact time with the control specimen. 

Percent reduction calculation:(4)$$\mathrm{Percent}\;\mathrm{Reduction}=1-10^{-\mathrm{Mv}}$$

#### 2.6.3. Anti-Fungal Activity

Control and test specimen of 2” × 2” dimension was cut for the testing. Fungal species were grown separately on Sabouraud dextrose agar for 7–14 days. The spore suspension of the fungi was prepared by pouring 10 mL of sterile DI water containing 0.5 mL of Tween 20 into the culture plate. The surface growth was gently scraped from the culture. The spore suspension was transferred into a centrifuge tube containing 25 mL of sterile DI water. The centrifuge tube was vortexed for one minute to break the spore clumps. The spore suspension was filtered to remove mycelial fragments. The spore suspension was washed three times in DI water by centrifugation and diluted to achieve a 1.0 × 10^6^ spore/mL for each fungal species. Spore suspensions were then combined using equal volumes of resultant spore suspension. Both test sample and control were placed separately onto Sabouraud dextrose agar, and an even layer of spore suspension was sprayed onto each material sample. Plates were incubated at 29 °C ± 1 °C and examined weekly for 28 days (Relative humidity >85%). All tests were performed in triplicate. The PLA filter media were tested against *Aspergillus niger* (AATCC 16888) following ASTM G21 standard [45].

### 2.7. Washability of the Filter Media

Laundering of the PLA NF filter media were performed in accordance with ISO 6330 [49] standard using the following test parameters: 3G:30 °C, gentle setting (wool/silk/synthetics). The washed PLA NFs filter media were tested for filtration performance following ASTM F3502 [41] protocol and tested against *Staphylococcus aureus* and *Klebsiella pneumoniae* following test method outlined in ISO 20743 standard [43].

## 3. Results and Discussion

NanoLayr Ltd., based in Auckland, New Zealand, is a producer of large-scale advanced nanofiber textiles certified AS9100d. We have recently launched a unique filter media marketed as FilterLayr^TM^ Eco. The product has a three-layer structure which consists of a middle layer composed of PLA NFs and manuka triketone at an average areal weight of 2.0 ± 0.1 gsm sandwiched between two outer layers of PLA spun-bonded non-woven fabric (Figure 1). Figure 1B,C displays SEM images of NFs produced from PLA solutions, randomly oriented and with an average fibre diameter of 168.3 ± 43.6 nm. The NFs display the typical stacked layered morphology of non-woven fibrous fabrics. The implementation of nano-scale fibres has been shown to be advantageous in air filtration due to the small diameter of the fibres and the corresponding high surface-to-volume ratio, enhancing the capture of particles through interception and other mechanisms [11,19,50,51]. Furthermore, homogeneous porosity can lead to lower pressure drop due to slip flow effects, made possible by the small fibre diameter when compared to microfiber counterparts [50]. 

### 3.1. Filtration Performance of PLA NF Filter Media

In a previous work, the degree of NF uniformity has been demonstrated by plotting the pressure drop of the NF filter media against the areal weight of NF material, showing linearity [11]. In an idealised situation, an increase in pressure drop will result in a proportional increase in filtration efficiency of the material. Figure 2 shows the relationship between the filtration efficiency and the pressure drop of the filter media when tested at three different air velocities as determined by international test standards (A) ASTM F2299 [40], (B) NIOSH 42CFR84 [42], and (C) ASTM F3502 [41]. All the filter media showed a linear relationship between pressure drop and filtration efficiency, with R^2^ values ranging from 0.848 to 0.979 depending on the test standard.

Up to 13 individual samples of the PLA NF filter media were tested according to the three international standards. Filtration efficiency and pressure drop values are displayed in Table 4. Filter media samples were challenged with monodispersed polystyrene latex sphere (PSL) aerosols with an average particle diameter of 0.1 µm (Figure 2A) and with NaCl aerosol with an average particle diameter of 0.3 µm (Figure 2B,C). The PLSs were nebulised, dried, and passed through the filter media at an airflow velocity of 28.3 L·min^−1^, as required for the PFE test method in ASTM F2299 [40]. All filter media samples containing PLA NF layers met the particulate filtration requirement for level 2 as specified in ASTM F2299 standard [40]. It displayed filtration efficiencies ≥98% (Figure 2A) and relatively low pressure drops ranging from 48 to 59 Pa (air velocities = 8 L·min^−1^). In addition, samples of the same areal weight were sent to a third-party-certified laboratory to ensure the efficacy of PLA NF filter media to pass the requirements of the test method ASTM 2100 [51]. The PFE, VFE and BFE results are presented in the supporting document (Appendix A). The filter media exceeded the level 2 criteria outlined in the ASTM F2100 standard [51] while following the efficiency values obtained for the various tests, PFE results above 99.5% (Appendix A), VFE results above 99.9% (Appendix A), and BFE results above 99% (Appendix A), with an airflow resistance requirement below 58.8 Pa (Appendix A). 

The PLA filter media were also challenged with NaCl aerosol with a count median diameter (CMD) of 75 ± 20 nm and a geometric standard deviation (GSD) of 1.86, equivalent to an average particle diameter of 0.3 µm (Figure 2B,C) which is used for both NIOSH 42CFR84 N95 [42] and ASTM F3502 test methods [41]. The NaCl particles were nebulised, dried, and passed through the filter media at an airflow velocity of 60 or 85 L·min^−1^ as required for the PFE test method in ASTM F3502 [41], and NIOSH 42CFR84 N95 [42], respectively. The PLA filter media exceeded the NIOSH 42CFR84 N95 particle filtration efficiency requirements of ≥95% as well as meeting the N95 inhalation and exhalation requirements of <314 and <245 Pa at 85 and 120 L·min^−1^, respectively. 

In the standard for barrier face coverings, ASTM F3502 [41], a minimum filtration efficiency of 20% is necessary to meet level 1, and ≥50% filtration efficiency is required to meet level 2. All PLA NF filter media samples tested exceeded the filtration performance requirements of ASTM F3502 level 2 [41]. Up to 98% filtration efficiency was achieved by the PLA NF filter media. Each of the samples tested in accordance with ASTM F3502 passed the level 1 breathability requirements. The PLA filter media were also tested in a certified laboratory to ensure their ability to pass the requirements of ASTM F3502 [41] and NIOSH 42CFR84 N95 [42]. The reports from the certified laboratory are presented in the supporting document (Appendix A). 

To verify the reusability of the filter media, the samples were subjected to 10 laundering cycles and then challenged with NaCl particles, as described above. The results showed a slight decrease in filtration efficiency (Appendix A). Out of the 10 samples that were laundered, one was damaged during the laundering cycles, and therefore was not tested. The remaining nine samples that were tested achieved an average filtration efficiency of 58.97% which readily meets the minimum requirement for level 2 filtration efficiency, in accordance with ASTM F3502 standard [41]. One of the nine samples that were tested is also suspected to have suffered some damage during laundering which consequently affected the overall average result.

### 3.2. Antimicrobial Activity of PLA Nanofibers Containing Manuka Triketone

In this work, natural and non-toxic manuka oil 5% wt. has been used to generate antimicrobial nanofibre filter media. The antibacterial properties of the filter media were tested according to ISO 20743:2013 [43] standard using three different bacteria: ATCC 6538P (*Staphylococcus aureus*), ATCC 4352 (*Klebsiella pneumoniae*) and ATCC 8739 (*Escherichia coli*) (Figure 3, left and Table 5). The electrospun PLA filter media were challenged against the test standard ISO 18184:2019 [44] using three different viruses: *influenza A* [H1N1] (ATCC VR-1469), human coronavirus 229E (ATCC VR-740) and SARS-CoV-2 (Delta variant B.1.617.2; NCBI MZ574052) (Figure 3, right and Table 6). Additionally, antifungal activity was tested according to ASTM G21 using ATCC 16888 (*Aspergillus niger*) (Table 7) [45]. 

#### 3.2.1. Antibacterial Activity of PLA Nanofibers Containing Manuka Triketone

The antibacterial activity of the filter media were tested according to ISO 20743:2013 [43]. The antibacterial test results confirmed that the manuka oil was integrated into the PLA NF mat and imparted excellent biocidal action against both Gram-negative and Gram-positive bacteria. The nanofibers loaded with manuka oil showed bacterial population reductions of 99.999% against *S. aureus*, 99.96% against *E. coli.* and 99.82% against *K. pneumoniae strains*, after a contact time of 18 h (Figure 3 and Table 5).

In order to test the retention of antimicrobial activity of the filter media after multiple washes, the samples were subjected to 5 and 10 laundering cycles, and the antibacterial performance was re-evaluated (Table 5). The laundering step was carried out in accordance with ISO 6330:2013 [49] standard using the following test parameters: 3G:30 °C, gentle setting (wool/silk/synthetics). Both the Gram-positive (ATCC 6538P) and Gram-negative (AATCC 4352) bacteria were used to challenge the PLA NF filter media. The sample showed excellent antibacterial activity even after 5 and 10 laundering cycles. The reduction rates of *Staphylococcus aureus* and *Klebsiella pneumoniae* bacteria after five washes upon a contact time of 18 h were 99.996% and 99.834%, respectively. After 10 washes, the reduction rates of *Staphylococcus aureus* and *Klebsiella pneumoniae* after a contact time of 18 h were 99.993% and 99.678%, respectively. 

#### 3.2.2. Antiviral Activity of PLA Nanofibers Containing Manuka Triketone

The antiviral results of the filter media were tested according to ISO 18184:2019 [44]. The results of the tests showed that the manuka oil embedded into the PLA nanofibers has outstanding virucidal properties against influenza A (H1N1), human coronavirus 229E and SARS-CoV-2 (Delta variant) (Figure 3, right and Table 6). The PLA filter media showed a virus reduction of 99% against influenza A, 99.82% against human coronavirus 229E and 99.69% against SARS-CoV-2 after a contact time of 2 h.

#### 3.2.3. Antifungal Activity of PLA Nanofibers Containing Manuka Triketone

The antifungal/fungicidal activity of the PLA NF filter media were evaluated using the following test method: ASTM G21—Resistance of Synthetic Polymeric Materials to Fungi (Table 7) [45]. The filter media displayed antifungal property and showed no *Aspergillus niger* growth after a period of 28 days (Table 7). 

## 4. Conclusions

Novel antimicrobial air filtration media (marketed as FilterLayr^TM^ Eco) containing electrospun PLA NFs loaded with manuka triketone have been developed by NanoLayr Ltd. FilterLayr^TM^ Eco has been proven to have excellent particle filtration efficiency and breathability and meets the requirements for following standard certifications, NIOSH N95, ASTM F2100 level 2, and ASTM F3502 level 1 breathing resistance and level 2 filtration efficiency. In addition to this, the filter media successfully passed ASTM F3502 level 2 filtration efficiency after 10 laundering cycles, making it reusable for a minimum of 10 washes.

FilterLayr^TM^ Eco is unique in its ability to both trap and kill airborne bacteria and viruses using naturally occurring antimicrobial Manuka oil as the biocidal agent. This enables the filter media to self-decontaminate and allows it to be used multiple times before it needs to be replaced, as it does not harbour or proliferate the microbes that it captures. This study demonstrated the effectiveness of the PLA filter media in neutralising different types of bacteria, viruses, and a fungus, namely, *Staphylococcus aureus*, *Klebsiella pneumoniae*, *Escherichia coli*, influenza A, human coronavirus 229E, SARS-CoV-2-Delta variant and *Aspergillus niger*. The filter media has also shown excellent performance against bacteria when tested after 5 and 10 laundering cycles, thus proving its ability to retain its anti-microbial properties for a minimum of 10 wash cycles, making it reusable and more sustainable. FilterLayr^TM^ Eco is also made from industrially compostable polymers, thus reducing the amount of waste being sent to landfills and consequently reducing plastic waste in the environment.

## Figures and Tables

**Figure 1 membranes-12-00571-f001:**
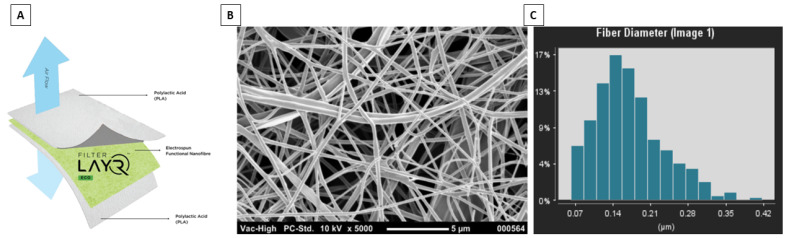
(**A**) Diagram of FilterLayr^TM^ Eco structure spun-bond PLA/PLA electrospun nanofibre/spun-bond PLA; (**B**) Scanning electron micrograph of nanofibre layer made from PLA; and (**C**) average fibre distribution of PLA electrospun fibres using Fibraquant image analysis software.

**Figure 2 membranes-12-00571-f002:**
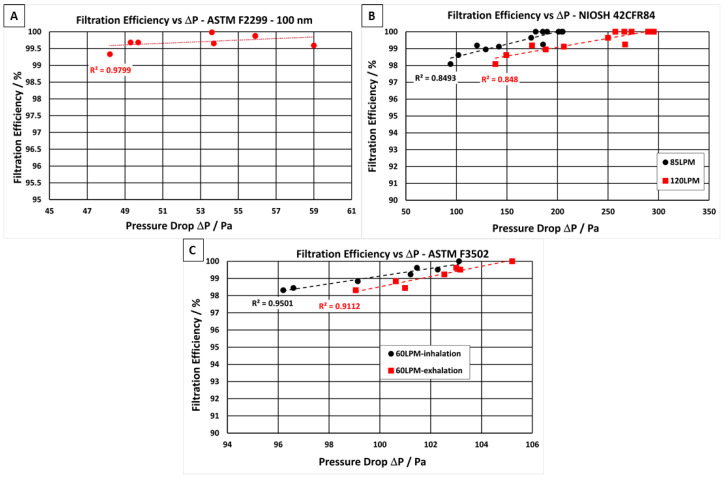
Filtration efficiency vs. pressure drop for PLA NF filter media tested in accordance with the following international standards: (**A**) ASTM F2299 [40], (**B**) NIOSH 42CFR84 [42] and (**C**) ASTM F3502 [41].

**Figure 3 membranes-12-00571-f003:**
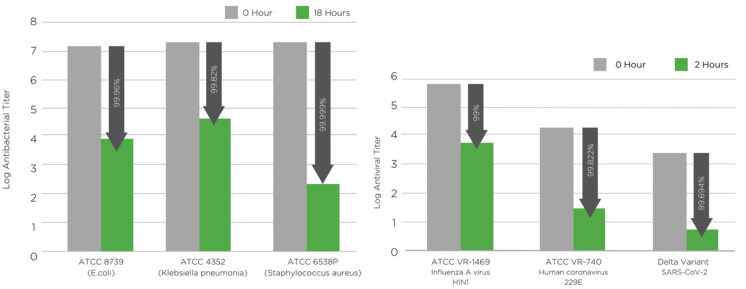
(**left**) Antibacterial activity of PLA NF filter media containing manuka triketone (5 wt.%) against *S. aureus* (ATCC 6538P), *E. coli* (ATCC 8739) and *K. pneumoniae* (ATCC 4352) tested according to ISO 20743:2013 [43]. (**right**) Antiviral activity of PLA NF filter media containing manuka triketone (5 wt.%) against influenza A, human coronavirus 229E and SARS-CoV-2 tested according to ISO 18184:2019 [44]. The arrows point at the inhibition of microbial growth.

**Table 1 membranes-12-00571-t001:** Filtration test method requirements.

Test		Level 1	Level 2	Level 3
ASTM F2299 (PFE) Filtration at 0.1 µm—28.3 L·min^−1^		95%≤	98%≤	
EN14683 (Breathing resistance—Breathability)		≤49 Pa	≤58.8 Pa	
ASTM F2101 -19 (BFE) Filtration at 3 µm—28.3 L·min^−1^		95%≤	98%≤	
ASTM F2101 (VFE) Filtration at 3 µm—28.3 L·min^−1^		95%≤	98%≤	
ASTM F3502 Filtration at 0.3 µm—60 L/min		20% £	50%≤	-
ASTM F3502 (Breathing resistance—Breathability)		15 mmH_2_O (147.5 Pa)	5 mmH_2_O (49 Pa)	-
NIOSH 42 CFR 84 Filtration at 0.3 µm—85 L·min^−1^		N95 95%≤		
Breathing resistance	Inhalation—120 L·min^−1^	<314 Pa		ΔP < 98 Pa
	Exhalation—85 L·min^−1^	<245 Pa		

**Table 2 membranes-12-00571-t002:** Antibacterial activity rating associated with percentage reduction of bacterial growth.

Antibacterial Activity Value (A)	Efficacy Rating	A Value	Reduction (%)
2≤A<3	Good Effect Level	1	90
		2	99
		3	99.9
A≥3	Excellent Effect Level	4	99.99
		5	99.999

**Table 3 membranes-12-00571-t003:** Antiviral activity rating associated with percentage reduction of virus growth.

Antiviral Activity Value (Mv)	Efficacy Rating	Mv Value	Reduction (%)
2≤Mv<3	Good Effect Level	1	90
		2	99
		3	99.9
Mv≥3	Excellent Effect Level	4	99.9
		5	99.999

**Table 4 membranes-12-00571-t004:** Pressure drop and filtration efficiency results for PLA NF filter media tested in accordance with ASTM F2299 [40], ASTM F3502 [41] and NIOSH 42CFR84 [42].

	ASTM F2299		ASTM F3502			NIOSH 42CFR84		
	Filtration Efficiency at 100 nm	ΔP	Filtration Efficiency at 300 nm	ΔP Inh. at 60 LPM	ΔP Exh. at 60 LPM	Filtration Efficiency at 300 nm	ΔP Ex. at 85 LPM	ΔP Inh. at 120 LPM
1	99.87	55.90	98.31	96.20	99.05	98.08	94.20	138.58
2	99.88	55.90	98.44	96.60	100.99	98.61	102.00	149.37
3	99.59	59.00	98.83	99.13	100.63	99.18	120.47	174.84
4	99.65	53.70	99.23	101.21	102.54	98.95	129.13	188.36
5	99.68	49.30	99.62	101.46	103.01	99.11	142.08	206.31
6	99.33	48.20	99.51	102.28	103.16	99.64	174.00	250
7	99.98	53.60	100	103.12	105.21	100	178.44	257.3
8	99.68	49.70				100	185.48	266.18
9						99.24	185.77	266.95
10						100	189.76	273.38
11						100	201.14	289.63
12						100	204.18	295.12
13						100	205.14	295.33

**Table 5 membranes-12-00571-t005:** Antibacterial reduction rate of PLA NF filter media containing manuka triketone (5 wt.%) against *S. aureus* (ATCC 6538P), *E. coli* (ATCC 8739). and *K. pneumoniae* (ATCC 4352) tested according to ISO 20743:2013 [43]. Reusability following 5 and 10 laundering cycle according to ISO 6330:2013 [49] and ISO 20743:2013, was confirmed for the PLA NF filter media when tested against *S. aureus* (ATCC 6538P) and *K. pneumoniae* (ATCC 4352).

ISO 20743:2013	*S. aureus* (ATCC 6538P)			*E. Coli* (ATCC 8739)	*K. pneumonia* (ATCC 4352)		
	0 Wash	5 Washes	10 Washes	0 Wash	0 Wash	5 Washes	10 Washes
Control	Initial			Initial	Initial	Initial	
Log CFU	4.69			4.69	4.83	4.80	
Contacting time (hours)	-			-	-	-	
Control	After contacting			After contacting	After contacting	After contacting	
Log CFU	7.25			7.19	7.3	7.6	
Contacting time (hours)	18			18	18	18	
Log CFU samples	2.33	2.87	3.1	3.83	4.55	4.82	5.11
Percentage reduction samples (%)	99.999	99.996	99.993	99.957	99.82	99.834	99.678
Log reduction samples	4.92	4.38	4.15	3.36	2.75	2.78	2.49

**Table 6 membranes-12-00571-t006:** Antiviral reduction rate of PLA NF filter media containing manuka triketone (5 wt.%) against influenza A, human coronavirus 229E and SARS-CoV-2 tested according to ISO 18184:2019 [44].

ISO 18184:2019	Influenza A (H1N1) (ATCC VR-1469)	Human Coronavirus 229E (ATCC VR-740)	SARS-CoV-2 (Delta Variant)
Infective titer test	TCID50 method	TCID50 method	Plaque assay
Log(Va) (Control, immediately)	5.88	4.25	3.21
Contacting time (hours)	2	2	2
Log(Vc) (Sample, after contacting)	3.88	1.5	0.70
Antiviral activity value, Mv	2	2.8	2.5
Percentage reduction samples (%)	99	99.82	99.69

**Table 7 membranes-12-00571-t007:** Antifungal activity of PLA NF filter media containing manuka triketone (5 wt.%) against *Aspergillus niger* tested according to ASTM G21 [45]. Note on antifungal rating: 0 no growth, 1 trace of growth (less than 10% coverage), 2 light growth (10–30% coverage), 3 medium growth (30–60% coverage) and 4 heavy growth (60–100% coverage).

Sub-Samples	Control Specimen	Test Specimen
Contact time	28 days	28 days
Rating	4	0
Observed Growth	High Growth	No Growth

## Data Availability

The data presented in this study are available on request from the corresponding author.

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
