# Peer review of "Virucidal and Bactericidal Filtration Media from Electrospun Polylactic Acid Nanofibres Capable of Protecting against COVID-19"

_membranes, 2022, doi:10.3390/membranes12060571_

Round 1

Reviewer 1 Report

The manuscript is well detailed and the experimental schemes are nicely elaborated. The authors have summarized the test standards nicely and it will help many researchers. The article can be accepted after some correction-

  1. Table 1 and Table 4 contradict with inhalation and exhalation flow rates.
  2. Figure 2- why there is a fitting line? Does it tell you anything?
  3.  The pressure drop increment is not linear across filters in different flow rates. Can authors elaborate the same? It may open a new discussion apart from facemasks.
  4. Page 8, line 291- what is the SD of the salt particles? Salt particles are highly polydisperse and can produce very wide range particle sizes.
  5. Mention quality factor and compare with some existing recent results.

Author Response

Thank you very much for your comment and feedbacks, please see answers below:

1. Table 1 and Table 4 contradict with inhalation and exhalation flow rates.

Table 4 has been updated, typographic mistakes

2. Figure 2- why there is a fitting line? Does it tell you anything?

It has been added to show an overall trend line. Authors forgot to comment on the R2 value. Comment on the trendline has been added in the section 3.1.

3. The pressure drop increment is not linear across filters in different flow rates. Can authors elaborate the same? It may open a new discussion apart from facemasks.

The pressure drop is not linear as there is a variability of areal weight with an average of 2.0 ± 0.1 gsm. The airflow used for each standard are different this is why the linearity differ between A, B and C in figure 2. 

4. Page 8, line 291- what is the SD of the salt particles? Salt particles are highly polydisperse and can produce very wide range particle sizes.

Sentence has been added.

5. Mention quality factor and compare with some existing recent results.

Authors are not interested in discussing quality factor in this study. This is outside of the scope of work. The paper present the performance of the filter media against international test standards which does not mention quality factor in their guideline.

Reviewer 2 Report

The subject of the paper entitled „Virucidal and Bactericidal Filtration Media from Electrospun 2 Polylactic Acid Nanofibre Capable of Protecting Against 3 COVID-19” written by Karabulut et al is very actual considering the challenges arising from the Covid 19 pandemic. 

Overall, this manuscript is written well, and the sections follow clearly. The conclusions are supported by the analysis of the results presented in the manuscript. 

Author Response

Thank you very much for your feedback

Reviewer 3 Report

This paper presents the usage of an active layer of mask as multifunction biomedical filter such as anti-viral and anti-bacterial active layer.
The topic is an interesting one and hot in the current time, however the paper is presented as a report rather than a research article which has serious flows which require essential modifications as follows:
1- There are no clear information about the solution chemical preparation and the used parameters of electrospinning.
2- The antibacterial part is missing more characterizations such as inhibition zone photos.
3- The filtration technique is missing more physical explanations; what is the main technique of anti-viral/anti-bacterial? Is it the blockage only?
4- More comparison between the behavior of your filtration layer and other commercial/literature layers is still needed.
5- Porosity analysis must be presented to support the breathability analysis.
6- The results of washability is not mentioned.

Author Response

Thank you very much for your comment and feedback. Please find below the answers:

1- There are no clear information about the solution chemical preparation and the used parameters of electrospinning.

The information relative to the solution is consider as trade secret and can not be revealed

2- The antibacterial part is missing more characterizations such as inhibition zone photos.

It has been carried out by third party laboratories and did not send inhibition zones.

3- The filtration technique is missing more physical explanations; what is the main technique of anti-viral/anti-bacterial? Is it the blockage only?

As mentioned, this filter layer uses Manuka oil (triketone) as an active layer. Manuka oil is known for its microbial properties but has never been incorporated into a nanofibre for filtration application.

4- More comparison between the behavior of your filtration layer and other commercial/literature layers is still needed.

As far as we know it is a unique product. It is commercially available product and must be compared with international test standard. Usually, academic test filter media at different airflow and not as the same exact condition as the international standard so the comparison of behavior will be inexistant.

5- Porosity analysis must be presented to support the breathability analysis.

R2 value is ranging from 0.848 to 0.979 which shows uniformity of the nanofibre.

6- The result of washability is not mentioned.

Table S7 in supplementary document shows the filtration efficiency of the filter media after washing (10 laundering cycles)

Table 5 in paper shows the antibacterial properties of the filter media after washing at 5 and 10 washes.

Round 2

Reviewer 3 Report

All comments have been addressed. Table of washability effect is preferred to be in the manuscript rather than supplemental file.